# Considerations and Technical Pitfalls in the Employment of the MTT Assay to Evaluate Photosensitizers for Photodynamic Therapy

Edith Alejandra Carreño [1,2], Anael Viana Pinto Alberto [1], Cristina Alves Magalhães de Souza [1], Heber Lopes de Mello [1], Andrea Henriques-Pons [3] and Luiz Anastacio Alves [1,*]

1   Cellular Communication Laboratory, Oswaldo Cruz Institute, Oswaldo Cruz Foundation FIOCRUZ, Av., Rio de Janeiro 4365, Brazil; acm.alejandra@gmail.com (E.A.C.); anael@ioc.fiocruz.br (A.V.P.A.); souzacam@ioc.fiocruz.br (C.A.M.d.S.); heberlopes07@gmail.com (H.L.d.M.)
2   School of Health Sciences, National Open and Distance University UNAD, Calle 14 sur # 14-23, Bogota 110311, Colombia
3   Laboratory of Innovations in Therapies, Teaching, and Bioproducts, Oswaldo Cruz Institute, Oswaldo Cruz Foundation FIOCRUZ, Av., Rio de Janeiro 4365, Brazil; andreah@ioc.fiocruz.br
*   Correspondence: alveslaa@gmail.com; Tel.: +55-21-25621809

**Abstract:** Photodynamic therapy (PDT) combines light, a photosensitizing chemical substance, and molecular oxygen to elicit cell death and is employed in the treatment of a variety of diseases, including cancer. The development of PDT treatment strategies requires in vitro assays to develop new photosensitizers. One such assay is the MTT 3-(4,5-dimethylthiazol-2-yl)-2,5-diphenyltetrazolium bromide developed in 1983 and widely used in PDT studies. Despite the exponential growth in the number of publications, a uniform MTT protocol for use in the PDT area is lacking. Herein, we list and standardize the conditions to evaluate the photosensitizer methylene blue (MB) in glioblastoma and neuroblastoma cell lines. In addition, we review technical pitfalls and identify several variables that must be taken into consideration in order to provide accurate results with MTT. We conclude that for each cell line we must have a dose-response curve using the MTT assay and good controls for the standardization. Additionally, the optimal values of the time and cell density must be in the linear range of the curve to avoid errors. We describe all relevant points and outline the best normalization techniques to observe the differences between treatments.

**Keywords:** MTT assay; cell MTT metabolic activity; photodynamic therapy

## 1. Introduction

Photodynamic therapy (PDT) is employed to treat a variety of skin diseases (such as actinic keratosis, basal cell carcinoma, and psoriatic plaque) [1–4] cancers (of the lung, bladder, colon, prostate, gastrointestinal, pancreatic, and breast), and fungal infections [5–11]. In PDT, exposure to light of the appropriate wavelength (400–700 nm) alters the conformation of a photosensitizer molecule, generating radicals and/or reactive oxygen species (ROS) that destroy the target cells [12–15]. The toxicity of the photosensitizer in the absence of light emission ("dark toxicity") should be negligible, and the light energy required to activate it should not damage neighboring healthy tissue. Preventing damage to non-target cells, either by the photosensitizer directly or by the reactive species [16,17], whilst improving drug efficacy, drives the development of new photosensitizers [18–20]. A key part of this research is in vitro toxicity tests with mammalian cell cultures.

One of the first assays developed to measure cell viability was the clonogenic assay, in which the ability of plated cells to proliferate is measured by counting the number of colonies formed after 1 to 3 weeks [21]. Over the years, several other assays, such as the MTT, XTT, WST-1, MTS, and Alamar Blue assays [22,23], have been developed, assessing functions required for cell survival, including cell membrane permeability, enzyme activity,

co-enzyme production, cell adherence, ATP production, and nucleotide uptake activity (Table 1) [21]. The most suitable assay depends on the type and quantity of cells, on the mechanism of cell death induced by the photosensitizer [24,25], and also on the cost and complexity of each test.

**Table 1.** Intracellular metabolic assays to measure cellular proliferation, cell viability, and cytotoxicity: their advantages and disadvantages.

| Name | Methodological Mechanism's Summary | Detection Method | Advantage | Disadvantage | Ref. |
|---|---|---|---|---|---|
| MTT Assay | MTT (2-(4,5-dimethyl-2-thiazolyl)-3,5-diphenyl-2H-tetrazolium bromide), a yellow tetrazole, is reduced to purple formazan in living and metabolic active cells through an enzymatic reaction | Spectrophotometer/ 570 nm | -Fast protocol; -High throughput | -Overestimation of viability; -Final solubilization step; -Reducing compounds are identified to interfere with tetrazolium reduction assays | [26,27] |
| XTT Assay | Actively respiring cells convert the XTT (sodium 2,3-bis(2-methoxy-4-nitro-5-sulfophenyl)-5-[(phenylamino)-carbonyl]-2H-tetrazolium inner salt) to a water-soluble, orange colored formazan product | Spectrophotometer/ 490 nm | -High sensitivity; -Large dynamic range; -Water-soluble | -Endpoint assay; -Overestimation of viability | [28] |
| WST-1 Assay | WST-1 (sodium 5-(2,4-disulfophenyl)-2-(4-iodophenyl)-3-(4-nitrophenyl)-2H-tetrazolium inner salt) is cleaved to a soluble formazan by a complex cellular mechanism that occurs primarily at the cell surface | Spectrophotometer /420–480 nm | -Highest sensitivity; -Faster protocol | -Endpoint assay; -Overestimation of viability | [29] |
| MTS Assay | MTS (5-[3-(carboxymethoxy) phenyl]-3-(4,5-dimethyl-2-thiazolyl)-2-(4-sulfo-phenyl)-2H-tetrazolium inner salt) in the presence of phenazine methosulfate (PMS), produces a formazan product | Spectrophotometer/ 492–490 nm | -one-step MTT assay: reagent straight to cell culture without intermittent steps | -Colorimetric interference, the intermittent steps in the MTT assay could remove traces of colored compounds | [30] |
| Alamar Blue | Resazurin compound that gets reduced to resorufin and dihydroresorufin in viable cells | Fluoro/colorimetric 560 nm excitation/ 590 nm emission filter set | -Measurement in both fluorometric and colorimetric plate readers | -Necessary to test cross-reactivity, -Necessity of additional control of the assay to verify cross-reactivity with any compound to be tested in a well without cells | [31] |

Due to its low cost and ease of implementation, the MTT assay [26] has been extensively employed over the past 30 years to evaluate new PDTs, and its use continues to increase: Between 2006 and 2019 the number of publications found in Google Scholar using "MTT assay" and "Photodynamic Therapy" as search terms increased by a factor of 10, reaching more than 2000 in 2019 (Figure 1). Nonetheless, some important technical complexities and limitations can lead to inappropriate use of the MTT assay. For instance,

excess MTT salt concentration can lead to nonspecific cell death depending on the cell type [32] and cell density directly correlates with resistance to oxidative stress in vitro [33]. Moreover, the assays rely on the reduction of MTT, a cell-permeable yellow compound, to a purple (500–600 nm light-absorbing) formazan product by NADPH-dependent enzymes in the mitochondria of viable cells; but a variety of potential photosensitizing compounds can directly reduce MTT to formazan, affecting the reliability of the assay [34–38]. Finally, a method of normalizing MTT assay experimental data to allow for inter-laboratory comparability is lacking, despite its key importance for the discovery of new PDT photosensitizers. Therefore, in this study, we sought to standardize the MTT protocol for in vitro photodynamic assays. Using the MTT assay in glioblastoma (U87, GL261) and neuroblastoma (SHSY5Y) cell lines, we determined the dark cytotoxicity of the methylene blue (MB) photosensitizer and were able to identify several pitfalls that influenced our results. Also, parameters used in high throughput screening (HTS) assays were used to optimize the MTT assay, decreasing the signal-to-noise ratio. As a result, we describe here a standardized protocol for using the MTT assay by applying HTS that should increase reliability and reproducibility levels in photosensitizer discovery.

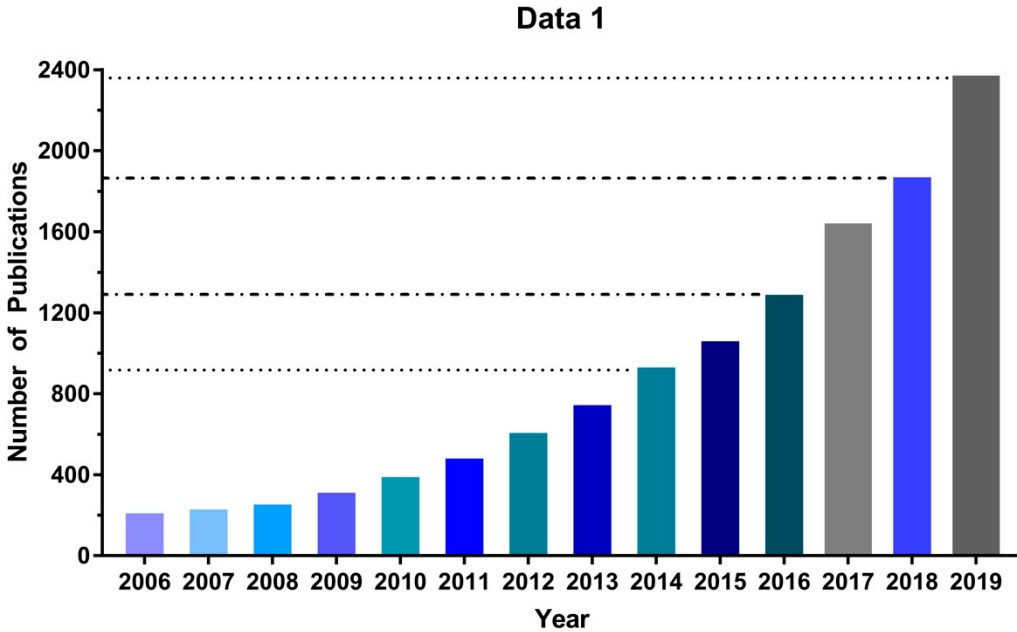

**Figure 1.** The number of publications per year using MTT assay in photodynamic therapy (PDT) studies. Source: Google Scholar, keywords: MTT assay "photodynamic therapy".

## 2. Materials and Methods

### 2.1. Cell Culture and Methylene Blue Photosensitizer Preparations

Glioblastoma cell lines U87 MG (ATCC® HTB-14™) and GL261 (ACC 802) and neuroblastoma cell line SH-SY5Y (ATCC® CRL-2266™) were cultured in DMEM-F12 and DMEM media (Sigma-Aldrich, São Paulo, Brazil) supplemented with 10% fetal bovine serum (FBS) (Gibco, Gaithersburg, MD, USA), 100 U/mL penicillin, and 100 μg/mL streptomycin (Gibco, Gaithersburg, MD, USA), according to ATCC recommendations and Kovalevic and Langford [39]. For all treatments, cells were detached with 0.025% trypsin and 0.4% ethylenediaminetetraacetic acid (EDTA) for 2 min, followed by inactivation with fetal bovine serum when 90% of the cells were detached from the bottles. Detached cells were seeded in 96-well plates (Corning, USA) at a density of $1 \times 10^4$/well and incubated for 48 h at 37 °C in 5% $CO_2$. The methylene blue (MB) photosensitizer (Merck, Darmstadt, Germany) was dissolved to 10 mM in ultrapure water (Purelab Ultra; ELGA, Woodridge, IL, USA) and stored at 4 °C and protected from light [40]. The cells were kindly provided

by the Laboratory of Immunopharmacology, Oswaldo Cruz Institute, FIOCRUZ (Rio de Janeiro, Brazil).

## 2.2. Cytotoxicity Assays

MTT assays were carried out according to Pacheco et al. [41], and all experiments were done immediately after PDT exposure except for the controls without PDT. For the assay, 25 μL of 5 mg/mL MTT in sterile PBS was added to each well ($1 \times 10^4$ cells/well). MTT specific activity was determined after 3 h incubation at 37 °C and 5% $CO_2$. Data are expressed as a percentage of viability when compared with untreated cells (negative control—considered as 100% of viability) and cells treated with Triton X-100 (positive control—considered as 100% of cell death). The controls should be performed to obtain a relatively suitable number of viable cells. Furthermore, the blank included during the MTT incubation was wells without cells (medium only) containing MTT solution for proper background subtraction (optical density (OD) measure in the experiment and control wells—OD cell-free wells). The MTT formazan crystals were dissolved in dimethyl sulfoxide (DMSO). The optical density (OD) in each well was determined with a microplate reader (Spectramax 190, Molecular Devices, San Jose, CA, USA) using an absorption spectrum of 570 nm (SoftMax® Pro GxP Software) [42]. This spectrum is proportional to the number of viable cells in each well.

For the normalization of the data, differences between the proposed methodologies for high throughput screening (HTS) assays were studied.

## 2.3. Intrinsic (Dark) Cytotoxicity Assay of Methylene Blue

Cells ($1 \times 10^4$/well) were incubated with different MB concentrations (from 1 to 400 μM for intrinsic cytotoxicity and 25 to 400 μM for interference studies of MB), in the absence of light, at 37 °C for 1 h to assess intrinsic MB toxicity. Cell viability was estimated using the MTT assay, as described above [27].

## 2.4. Photodynamic Action Test of Methylene Blue

Cells were incubated in 96-well plates at a density of $1 \times 10^4$ with different concentrations of MB (from 5 to 400 μM) and then exposed to red light (620–750 nm, 18–23 mW/cm$^2$) for 6 min. Cell viability was estimated using the MTT assay as described above [27]. For background subtraction, the blank included during the MTT incubation was cell-free wells with medium, containing only the MTT solution.

## 2.5. Statistical Analysis

The result of each experiment is shown as mean ± SD. Data are means of at least three independent experiments in triplicate. To test if the results fit a Gaussian distribution, the D'Agostino-Pearson normality test was adopted. If the data fit a Gaussian distribution, a parametric test (ANOVA) with Bonferroni post hoc test was used; if not, a nonparametric test (Kruskal–Wallis) with Dunn post hoc was considered. Differences were considered significant at *p* values < 0.05. Statistical and graphical analyses were performed using Graph Pad PRISM version 7.00 for Windows (GraphPad Software, San Diego, CA, USA).

For the standardization of the cell inoculum for the MTT reduction assay. Data are expressed as mean ± standard deviation. To test the homogeneity of variance, we used Bartlett's test. For the statistical comparison of the data, we used two-way ANOVA followed by Sidak's post-test. Representative graphs are of three independent experiments performed in triplicate.

*2.6. Z'factor Index*

The Z'factor is a statistical measure of the effect size and represents the signal window available for an assay in terms of total separation among the positive control (no treatment) and the negative control (cytotoxic treatment). Therefore, it is a useful method to describe an assay, and it has become standard, especially in HTS assays. The Z'factor was calculated according to:

$$Z' = 1 - \frac{3(\sigma_p + \sigma_n)}{|\mu_p - \mu_n|}$$

where $p$ is the $OD_{570}$ (optical density at 570 nm) of the positive control, n is the $OD_{570}$ of the negative control, $\sigma$ is the standard deviation of $OD_{570}$, and $\mu$ the mean of $OD_{570}$. Hereafter, we set the Z'factor according to Zhan et al., 1999. To acquire a well-defined signal window, the Z'factor must be greater than 0.5 and less than 1 (Table 2).

**Table 2.** Interpretation of the calculation of the Z'factor.

| Z-Factor | Interpretation |
|:---:|:---:|
| 1.0 | Ideal |
| Between 0.5 and 1.0 | Excellent assay |
| Between 0 and 0.5 | Marginal assay |

## 3. Results

*3.1. Cell Density for the MTT Reduction Assay in PDT*

A key question when using a viability test is whether cell density could influence the assay, as cells at lower density are more affected by the photosensitizer (PS) treatment [43]. Therefore, it is necessary to establish the appropriate number of cells/well to be used. In this study, this was tested taking, as a starting point, data from the literature reporting cell densities ranging from $2 \times 10^3$ to $4 \times 10^4$ for U87, $1 \times 10^3$ to $4 \times 10^4$ for SHSY5Y, and $4 \times 10^3$ to $2 \times 10^4$ for GL261 cell line.

Figure 2 shows the effect of increasing cell densities on $OD_{570}$ in an MTT protocol with or without incubation with Triton X-100 (0.4%). As cell density increased, so did the difference between treatments, in an S-shaped curve (insets, Figure 2A–C). For the SHSY5Y (Figure 2A) and U87 (Figure 2B) cell lines there was no difference between treatments ($p > 0.001$) at the lowest densities ($1 \times 10^3$ and $2 \times 10^3$, respectively), whereas for the GL261 cell line, Triton yielded a significantly higher absorbance at $4 \times 10^3$ cells/well. The effect of cell density saturated at $3 \times 10^4$, $2 \times 10^4$, and $1.2 \times 10^4$ cells/well, for the SHSY5Y, U87, and GL261 cell lines, respectively (Figure 2A–C).

Calculating the Z'factor for each cell line, we found that densities of $1–3 \times 10^4$ cells/well yielded the values closest to 1 h for all cell lines (Table 3). As a result, these densities were used in the PDT assays.

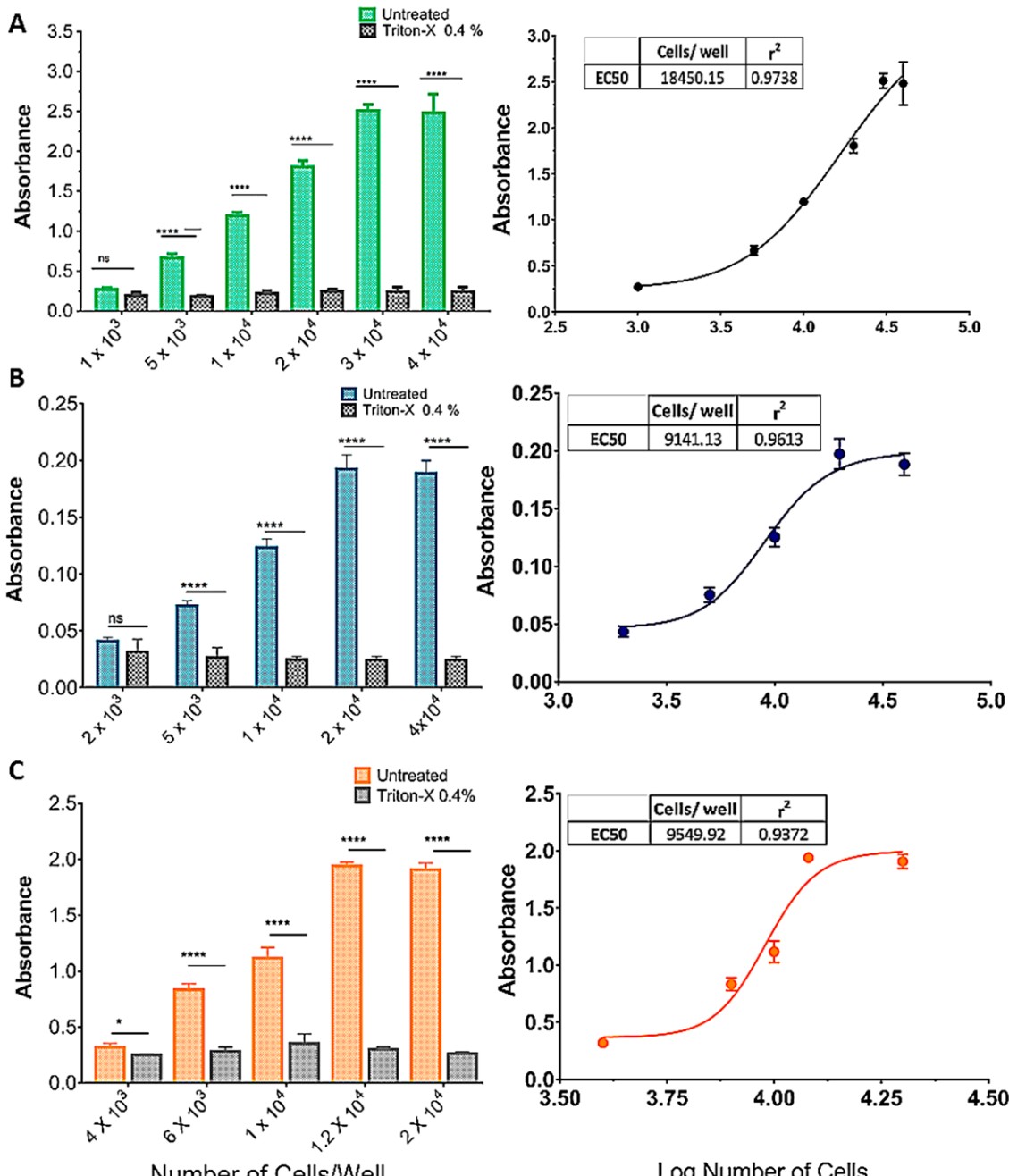

**Figure 2.** Standardization of the cell inoculum for the MTT reduction assay. (**A**) SHSY5Y, (**B**) U87, and (**C**) GL261 cell lines. The graphics show the effect of increasing cell densities on $OD_{570}$ in an MTT protocol with or without incubation with Triton X-100 (0.4%). As cell density increased, so did the difference between treatments, in an S-shaped curve (**A–C**). For the SHSY5Y (**A**) and U87 (**B**) cell lines, there was no difference between treatments ($p > 0.001$) at the lowest densities ($1 \times 10^3$ and $2 \times 10^3$, respectively), whereas for the GL261 cell line, Triton yielded a significantly higher absorbance at $4 \times 10^3$ cells/well. The effect of cell density was saturated for the SHSY5Y, U87, and GL261 cell lines at $3 \times 10^4$, $2 \times 10^4$, and $1.2 \times 10^4$ cells/well, respectively (**A–C**). Data are expressed as mean ± standard deviation. To test the homogeneity of variance, we used Bartlett's test. For the statistical comparison of the data, we used two-way ANOVA followed by Sidak's post-test ($p < 0.001$), ns = not significant, (* = $p < 0.05$; **** = $p < 0.001$). Graphs represent three independent experiments performed in triplicate.

**Table 3.** Z'factor values for SHSY5Y, U87, and GL261 cell lines.

| Cell Line | * Number of Cells | Z'Factor |
|---|---|---|
| SHSY5Y | 2000 | −0.4 |
| | 5000 | 0.7 |
| | 10,000 | 0.8 |
| | 20,000 | 0.8 |
| | 30,000 | 0.9 |
| | 40,000 | 0.7 |
| U87 | 2000 | −1.2 |
| | 5000 | 0.5 |
| | 10,000 | 0.7 |
| | 20,000 | 0.8 |
| | 40,000 | 0.8 |
| GL261 | 4000 | −0.7 |
| | 6000 | 0.6 |
| | 8000 | 0.5 |
| | 10,000 | 0.8 |
| | 12,000 | 0.9 |
| | 20,000 | 0.8 |

* Number of cells per well in a 96-well plate.

### 3.2. Viability Evaluation by the MTT Reduction Method

Depending on the experimental model, the MTT salt itself may induce significant cytotoxicity. For instance, after incubating U87 glioblastoma cells for 3 h (at 37 °C and 5% $CO_2$) with 0.75 mg/mL MTT (final concentration), as indicated in published protocols [44,45], we found 40% cell death as indicated by decreased $OD_{570}$ (data not shown). At this level of baseline viability, it would be impossible to accurately measure cytotoxicity by methylene blue. Therefore, to determine the optimal concentration of MTT, U87 cells, plated at $1 \times 10^4$ and maintained for 48 h at 37 °C and 5% $CO_2$, had their media replaced with media containing 0.125 to 0.500 mg/mL MTT for 1 h to 4 h and were then subjected to the MTT assay.

Figure 3 shows that 3 h is indeed the optimal incubation time for the assay, as $OD_{570}$ reaches a plateau after this time. Likewise, incubation with 0.375 mg/mL MTT resulted in the highest $OD_{570}$ at all time points. With 0.500 mg/mL MTT incubation, $OD_{570}$ was lower than with 0.375 mg/mL MTT. A cytotoxic effect of MTT might explain this result, as described elsewhere [27]. Thus, the concentration of MTT chosen for the viability assays was 0.325 mg/mL for all three cell lines, although the results were obtained with the U87 cell line, no adverse effects were observed with this concentration for the other cell lines.

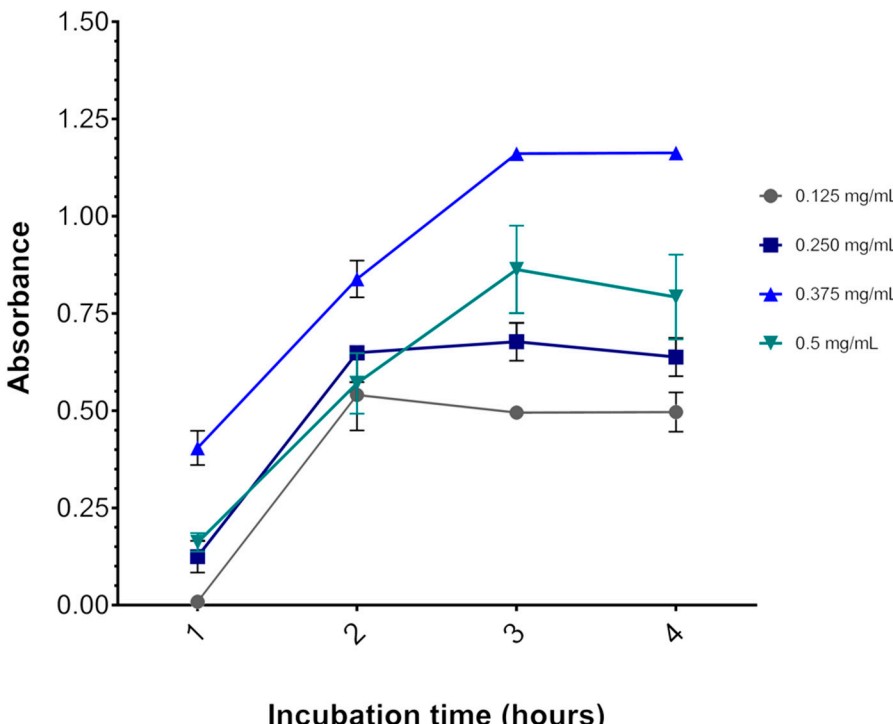

**Figure 3.** Effect of MTT concentration and incubation time on the U87 cell line. The graph represents the values of the absorbances as a function of incubation time. After 3 h of incubation at 37 °C, the absorbance reaches a plateau. The colored lines represent the MTT salt concentrations (gray = 0.125, dark blue = 0.250, light blue = 0.325, and green = 0.5 mg/mL MTT salt). Data are expressed as mean ± standard deviation. Representative graphs of three independent experiments performed in triplicate.

### 3.3. The Sensitivity of the Assay

Analytical sensitivity represents the smallest amount of substance in a sample that can accurately be measured by an assay. To determine the sensitivity of the MTT reduction assay, and considering that measured $OD_{570}$ is directly proportional to cell density, increasing amounts per well (or densities) of cells were plated to find the maximum cell density still in the linear range of OD, which should increase as cell density increases [46,47]. The assay was performed after 48 h of culture, and the absorbance data were plotted against the cell densities. As shown in Figure 2, the absorbance value increased with cell density and then reached a plateau at $3 \times 10^4$ cells for the SHSY5Y cell line, $2 \times 10^4$ cells for U87, and $1.2 \times 10^4$ for GL261.

### 3.4. Interference of Methylene Blue with the MTT Assay

The MTT assay is a colorimetric assay. Thus, a photosensitizer (PS) might interfere with the results if it is capable of absorbing electromagnetic energy in the same wavelength of the electromagnetic spectrum (570 nm) as the formazan product of the MTT reduction, as is the case for MB. To test for this, cells were treated for 1 h with MB concentrations ranging from 25 to 400 μM, then washed twice (Figure S1b) or not washed (Figure S1b) with PBS 1X before assessment of cell viability. Spectrophotometric measurements from 400 nm to 800 nm in 10 nm steps were performed. Our data showed that the presence of the MB did not interfere with the absorbance reading at 570 nm. Corroborating our data, Yong Kun Zou et al. (2016) showed that the maximum absorbance of MB can be seen at 664.5 nm and the MTT measurement was found at 570 nm [24].

### 3.5. Viability Assay and Data Normalization

To investigate MB dark toxicity and phototoxicity on the SHSY5Y, U87, and GL261 cell lines, those cells were incubated with 1 to 400 µM MB without light for 30 min at 37 °C and 5% $CO_2$. Half of them were then exposed to red light.

Given inter-experiment variability, the literature describes a variety of normalization methods to allow for comparison among results (Table 4). We compared these different methodologies, both those used with protocols for MTT assays [27] as well as some that were proposed for HTS assays [48–50]. Normalization methods, in general, produce high-quality response curves that allow a reliable characterization of cytotoxic effects while mitigating false positive or false negative test results; the chosen methodology was the one that best met this purpose.

**Table 4.** Normalization methods.

| Method | Equation | Observations | Reference |
|---|---|---|---|
| *Controls-based* | | | |
| Percent of control | $POC = \frac{X_i}{\hat{C}} \times 100\%$ | where *Xi* is the raw measurement of treatment and $\hat{C}$ is the mean of the measurements of the positive controls | [43,49] |
| % Maximum data Absorbance | $\%MAb = \frac{x_i}{MAb}$ | where *Xi* is the raw measurement of treatment and *MAb* is the maximum data absorbance of assay | W. R. |
| Percent of Median of control | $POMC = \frac{x_i}{M\ of\ control}$ | where *Xi* is the raw measurement of treatment and "*M of control*" is the median of the measurements of the positive controls | [50] |
| *Non-controls-based* | | | |
| Z score | $Z = \frac{(X_i - \hat{C})}{S_x}$ | where *Xi* is the raw measurement on the compound, $\hat{C}$ and *Sx* are the mean and the standard deviation, respectively, of all measurements within the plate | [51] |
| Percent of samples | $Z = \frac{X_i}{\hat{C}} \times 100\%$ | where *Xi* is the raw measurement on the compound and $\hat{C}$ is the mean of all measurements within the plate | [50] |
| Robust Percent of samples | $Z = \frac{X_i}{Med} \times 100\%$ | where *Xi* is the raw measurement of the compound and *Med* is the median of all measurements within the plate | [50] |

W. R. = without reference.

The normalization methods evaluated were separated into two sets: control-based normalization methods—CBN (Figure 4A–C) and non-control-based normalization methods—NCBN (Figure 4D–F).

All CBN methods showed the same differences between treatments, with a clear difference between positive (no treatment) and negative (0.4% Triton-X) controls. The NCBN methods (Figure 4D–F) do not require a positive or negative control to compare treatments as they use data of all the wells as an internal control; the samples are considered ineffective or inactive. Subsequently, these samples could be used as controls [51]. In the case of the Z-score, a D'Agostino–Pearson normality test was performed. It is important to mention that, in these types of analyses, we wanted to detect the outliers, data points not consistent with all other measurements [50]. Commonly, Z-scores with an absolute value of > 3 or < −3 are considered outliers from a normal distribution. The choice of 3 as the threshold for the absolute value (k = 3) originates the 3-sigma rule" [52].

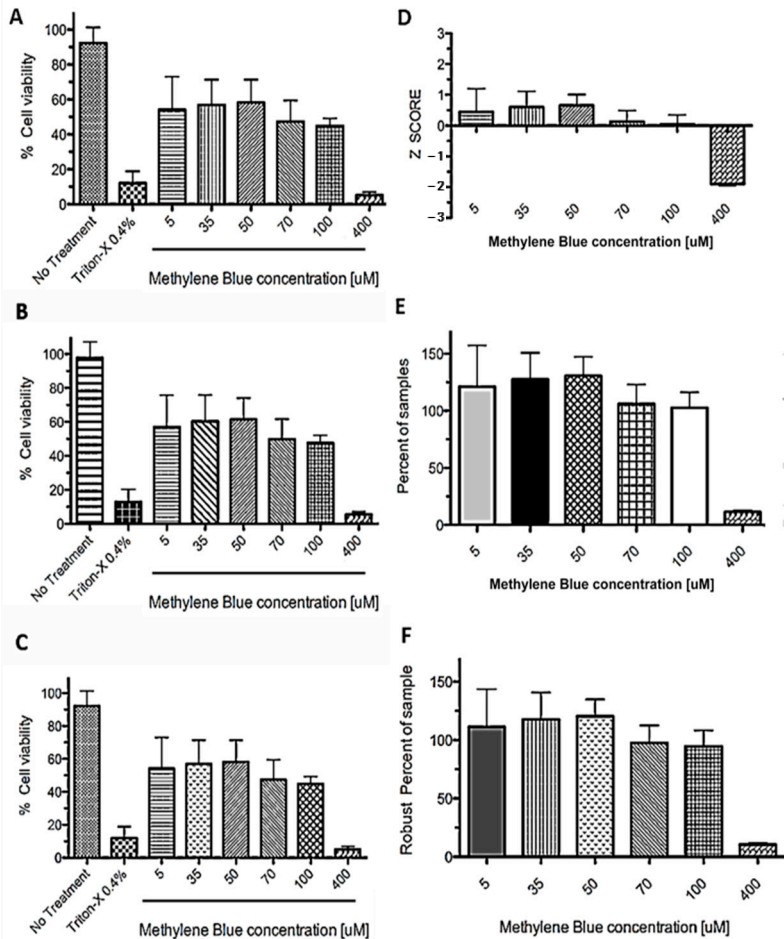

**Figure 4.** Data normalization methods applied to MTT assay after PDT. SHSY5Y cells were incubated in plates with different concentrations of methylene blue (MB) (from 5 to 400 µM) and then exposed to light for a 6 min treatment, then 0.75 mg/mL MTT was added (final concentration). The absorbance was measured after 3 h and the read in $OD_{570}$ values were normalized as: (**A**) Percent of control. (**B**) Percent of maximum $OD_{570}$. (**C**) Percent of the median of control. (**D**) Z score. (**E**) Percent of samples. (**F**) Robust percent of samples. Data are expressed as mean ± SD. Graphs are representative of three independent experiments performed in triplicate.

As seen in Figure 4, concentrations ranging from 5 to 100 µM show a similar distribution, with their z-scores between 0 and 1. However, a clear difference was observed for the concentration of 400 µM (z-score = −2). Although the 3sigma rule cannot be applied in this particular case, it is considered that at this concentration there is a significant effect on cell viability after photodynamic treatment. Therefore, statistical and graphical methods can be used to identify the outliers, allowing the identification of those photosensitizers with significant activity. In brief, when comparing the values of the effects of different concentrations of MB (5–400 µM) in the different analyses, pronounced cytotoxicity was observed only with 400 µM of MB. Similar results were found for the U87 cell line (data not shown).

On the other hand, the $EC_{50}$ curves exhibited a cytotoxic effect depending directly on the MB concentration (Figure 5). Consequently, higher doses of MB with the same time exposure of red light achieved significantly greater cytotoxic effects.

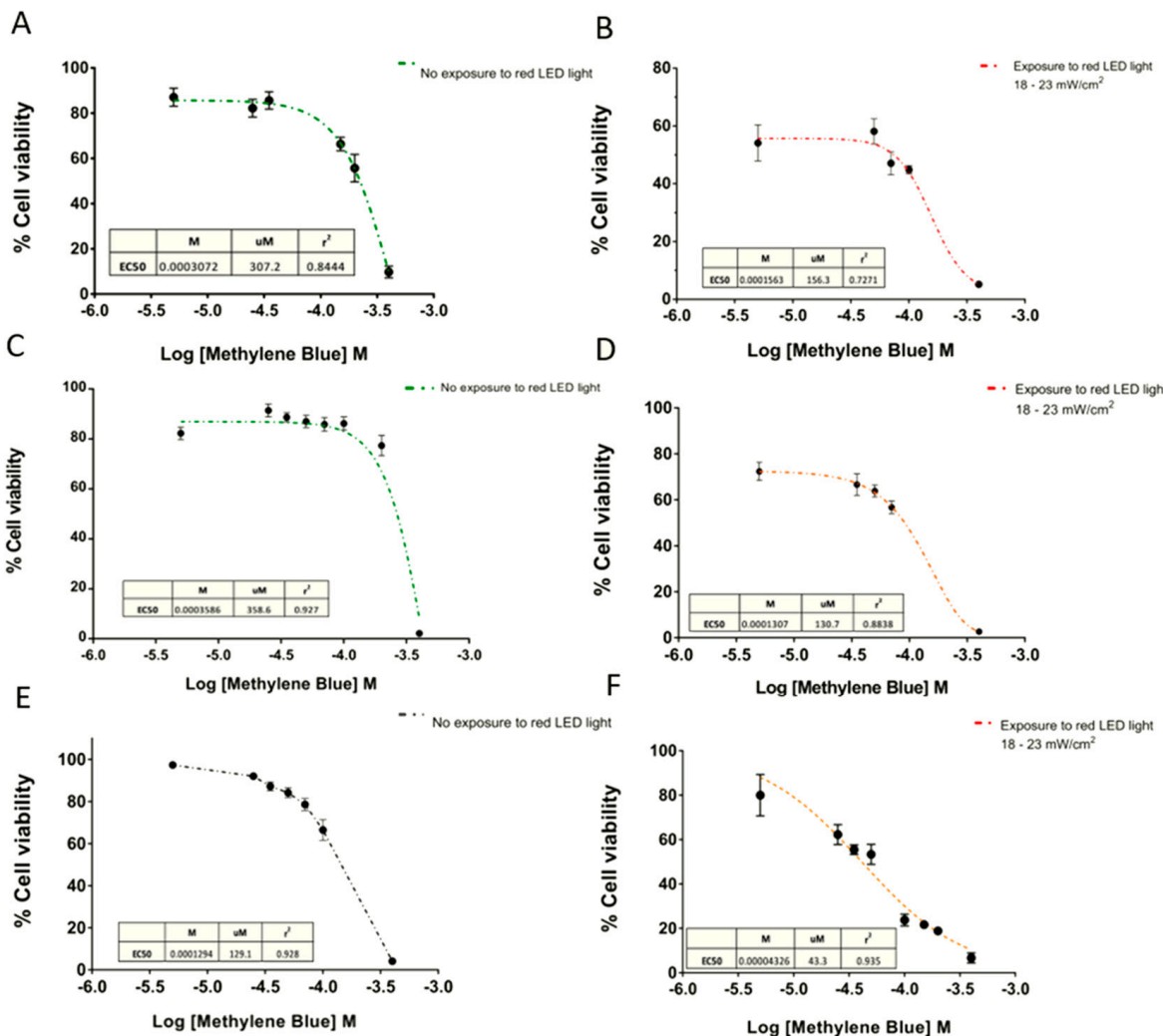

**Figure 5.** Dose-response curves of Methylene Blue. Dose-response curves in (**A** and **B**) SHSY5Y human neuroblastoma, (**C** and **D**) U87 glioblastoma, and (**E** and **F**) GL261 cell lines. The concentration of MB produced a 50% response of the maximal response (EC$_{50}$) calculated in absence of light, and after 6 min exposure at 18–23 mW/cm$^2$. Data are expressed as mean $\pm$ SEM. The data were normalized as a function of the mean negative control treatment (CN = cells without MB).

## 4. Discussion

This article aimed to identify key factors to be taken into account when designing in vitro PDT assays employing MTT as a cell viability test.

Cell density has been shown to modify the sensitivity of the cells to toxins, environmental stress, and extrinsic apoptotic signaling. For example, the cytotoxicity of ascorbic acid for malignant lymphoid CEM-C7 cells significantly increases when cell density decreases [53]. Kim, et al. [33], demonstrated that cell density directly correlates with resistance to oxidative stress in vitro. On the other hand, if cells get to the stationary phase, the OD$_{570}$ is not linear [47,54]. In our study, the OD$_{570}$ values decreased at the highest cell density, probably because cells had stopped dividing as a result of cell accumulation, shape change, and growth factor depletion. These results indicated that U87, SHSY5Y, and GL261 cells should be seeded at a density lower than $2 \times 10^4$/well and suggested that experiments using longer periods should be designed in such a way as to avoid an over confluence that could lead to depletion of nutrients from the culture medium and, therefore, lower cell viability. If the PS to be evaluated presents long-term toxicity, it is advisable to choose viability assays other than MTT [55].

The Z′-factor values for $1 \times 10^4$ to $2 \times 10^4$ cells/well were 0.8 and 0.9, respectively, which should provide a greater working range (positive and negative controls can be discriminated from each other), using the Z′factor calculations [52]. The advantage of using the Z′factor is the reduction of the amplitude and variability of the assay signal to only one parameter [56]. Several publications have described the use of Z′factor as a measure of quality control. For instance, Min et al. used Z′factor to demonstrate the utility of the SAMDI approach for chemical screening against protease activity of lethal anthrax factor [57]. In the same line, Hamid et al., compared the performance of Alamar blue and MTT cell viability assays in a high throughput format, reporting Z′-factor values of 0.85 and 0.82 for the Alamar blue and MTT assays, respectively. These values indicate that both assays are of high quality [58].

Regarding the conditions of the MTT assay, using the concentration described in the literature [42,59,60] we observed data fluctuation in some cases. MTT reduction can vary significantly between cell lines [61], and the formazan that results from MTT reduction can exhibit harmful effects, such as cell membrane damage during its exocytosis, inducing cell death [62]. Riss et al. [27], described morphological changes in Balb/c 3T3 cells during incubation with MTT solution at 0.8 mg/mL. Thus, we evaluated the effect of several MTT concentrations on the survival of the U87 cell line. Our results are consistent with a cytotoxic effect of MTT concentrations higher than 0.500 mg/mL (Figure 3).

In general, it is accepted that the MTT assay measures a set of enzymatic activities, which are related to cellular metabolism in different ways. The first description of MTT reduction by the mitochondrial electron transport chain was in the 1960s; since then, the mitochondrial II/succinate dehydrogenase complex has been erroneously seen as the unique site of intracellular reduction of MTT [63]. However, there is evidence of non-mitochondrial MTT reduction, through oxidoreductases, which use NADH/NADPH as cofactors. Superoxide may also contribute to the intracellular reduction of MTT, and cell surface electron transport is responsible for the extracellular reduction of MTT [64]. Moreover, glucose concentration and its absorption rate by the cells [61], the level of lactate, pyruvate, and NADH/NADPH [65], and fetal bovine serum (>10%) in the culture medium [66,67], have all been shown to influence MTT salt reduction as well.

One important point is that, throughout assay development and validation, microscopic observations, like cell detachment (in adherent cell lines) and the presence of cell fragments, are essential to confirm cellular toxicity. This is because the MTT assay can only distinguish viable cells from either senescent and dead cell, but not between senescent and dead cells. As senescent cells may continue transforming MTT to formazan, this could result in an assay artifact [58]. Thus, controlling these parameters through microscopic observations allows the MTT assay to provide a correct estimate of the number of viable cells.

*Viability Assay and Data Normalization*

Normalization of quantitative assay data is critical when interpreting effective biological system status to reduce fluctuations in the raw data between independent experiments and allow greater confidence in data comparisons [27,31,48]. Accordingly, we compared normalization methods (Table 4), which we divided into two categories: (1) controls-based (CB) methods, which, as the name suggests, assume that their controls are 100% or/and 0% of some effect—these methodologies have a significant problem when the controls do not work properly; (2) non-controls based (NCB) methods, which do not use comparisons with controls, but rather comparisons between samples, relying on the general distribution of values rather than depending on controls [51]. Some groups are using NCB normalization with the median, not the mean because this parameter is not affected by outliers.

The percent of control method is the most used in the MTT literature, and the other five normalization methods tested here provided similar results. Still, we suggest using at least one CB and one NCB test in MTT PDT assays to better visualize the data. Besides always making the experiments at least in triplicate will help evidence any outliers.

Photodynamic therapy (PDT) is a procedure based on the utilization of light-activated compounds or photosensitizers (PS). The rational discovery and design of PS molecules have been the subject of exhaustive research efforts of investigators [68]. Over the years, several techniques have been developed to screen candidate molecules, but most of these techniques are labor-intensive, time-consuming, produce waste products that are toxic or radioactive, and lastly, are expensive to perform [24,64]. In contrast, the MTT assay is a simple, low-cost procedure that can be adapted well to PDT protocols. In addition, toxicity is observed with the MTT assay at lower concentrations compared to the assays required for detection of toxicity with the lactate dehydrogenase leakage, the neutral red, and Alamar blue assays [51,65]. Ultimately, as mentioned, the MTT assay measures the metabolic activity of cells, including some mitochondrial and cytosolic dehydrogenases. Since apoptosis is a common response to PDT [27,42], this can be a reasonable method for detection of efficacy, as it will likely reflect the numbers of functional mitochondria and therefore survival. However, if other death pathways are involved, it can be inaccurate. With that in mind, MTT assay is just a quick test for phototoxicity that needs to be supplemented with clonogenicity appraisal. Clonogenic assays are frequently used to study the survival of irradiated cancer cells [43], though MTT assays are widely accepted to study chemosensitivity or toxicity of drugs in human tumor cell lines. In the literature, specific studies on the comparability of MTT and clonogenic assays with positive results can be found [44,45]. Equally relevant to consider is how the effects vary with the time between PDT treatment and carrying out the assay of MTT. For more precise results, the MTT test can be performed several hours (24 to 48 h after irradiation) after PDT treatment because you can have late repair or loss of toxicity.

Nevertheless, it is important to consider in PS screening that a variety of chemical compounds might affect the MTT assay result due to their ability to lead to a non-enzymatic reduction of the MTT to formazan, such as dithiothreitol [34], vitamin A, and ascorbic acid [35]. Moreover, the MTT assay should not be used in the presence of mitochondrial uncoupling agents, both natural and synthetic [36,69]. Plant extracts and polyphenolic compounds also have been reported to interfere with the MTT assay [37]. The use of a control without cells in the MTT assay has been proposed to avoid false-positives resulting from the presence of molecules with reducing properties, such as flavonoids, but evidence in the literature suggests that this may not be enough [70]. Moreover, a live/dead staining procedure in addition to the metabolic assay is recommended to validate the results, as well as the use of the appropriate normalization methods.

## 5. Conclusions

In vitro cytotoxicity assays are important in various fields of research. The assay should be sensitive enough to detect small differences in cell viability and reliable enough to generate reproducible results under different controlled experimental conditions.

We can conclude that, with careful standardization, it was possible to obtain ideal conditions for MTT assaying of cell viability after MB incubation and PDT. The underestimation of cell viability by the MTT assay and its significance depended on the cell line employed, a time point of viability measurement, and other experimental parameters. Furthermore, we provided a comprehensive examination of factors that should be taken into account when performing the MTT assay in PDT tests, and outlined how to normalize data to highlight the differences between samples.

**Supplementary Materials:** The following are available online at https://www.mdpi.com/2076-3417/11/6/2603/s1. Figure S1. Interference of Methylene Blue with the MTT Assay. OD spectra of 96-well plates of GL 261 cell lines ($1 \times 104$ cells/well) treated with Methylene Blue (MB) for one hour, then measured with a spectrometer. (A) MB-containing PBS was not washed. (B) MB containing PBS was washed with PBS. The red box indicates the 570 nm region of the spectra.

**Author Contributions:** Conceptualization, L.A.A.; methodology, E.A.C. and L.A.A.; software, E.A.C. and A.V.P.A.; validation, L.A.A.; formal analysis, E.A.C.; investigation, E.A.C.; resources, A.H.-P. and L.A.A.; data curation, C.A.M.d.S.; writing—original draft preparation, E.A.C.; writing—review and editing, A.V.P.A. and L.A.A.; visualization, C.A.M.d.S and H.L.d.M.; supervision, A.V.P.A.; project administration, L.A.A.; funding acquisition, L.A.A. All authors have read and agreed to the published version of the manuscript.

**Funding:** This research was supported, in part, by FIOCRUZ, Brazil.

**Acknowledgments:** This research was supported by FIOCRUZ, CNPq, and CAPES.

**Conflicts of Interest:** The authors have no conflict of interest to report.

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
