# Peer review of "Considerations and Technical Pitfalls in the Employment of the MTT Assay to Evaluate Photosensitizers for Photodynamic Therapy"

_applsci, doi:10.3390/app11062603_

Round 1
Reviewer 1 Report
The manuscript entitled “Considerations and Technical Pitfalls on Employing The Mtt Assay to Evaluate Photosensitizers for Photodynamic Therapy” by Carreño et al. reports on the standardization of the MTT protocol for a reliable utilization for the in vitro cytotoxicity studies of the photoactivatable compounds developed for photodynamic therapy. The studies were performed at the glioblastoma (U87, GL261) and neuroblastoma (SHSY5Y) cell lines with methylene blue used as the photosensitizer. Altogether, this work has clear scientific motivation, experiments are well-designed and the obtained results seem to be sound for the scientists working not only in the field of PDT compounds but in vitro cytotoxicity testing in general. I have only few comments, which should be answered in the revised text.
- section 2.3 – the background subtraction needs to be specified.
- section 3.1 – MTT instead of Mtt in the title
- line 170 – add “cells/well”, please.
- line 207 – 1 h
- line 216 – there is a discrepancy between the used concentrations of MB for the phototoxicity studies, as compared with the section 2.3.
Reviewer 2 Report
While this report does provide an analysis of assorted methods for assessing efficacy of PDT, there are issues not dealt with. True viability is best ascertained by clonogenic assays. The MTT assay measures the activity of some mitochondrial dehydrogenases, i.e., a test for mitochondrial damage. Since apoptosis is a common response to PDT, this can be a reasonable method for detection of efficacy, but if other death pathways are involved, it can be inaccurate.
There can also be repair or delayed toxicity which can be missed depending if the assay is carried out too soon after irradiation. Autophagy, or more particularly, mitophagy is a potential pathway for circumventing the ability of mitochondrial photodamage to induce cell death. If the assay is delayed for 24-48 hours after irradiation, then these concerns diminish. This needs to be discussed. In the procedure outlined, the MTT assay was carried out (line 88) immediately after irradiation,
Thus, this report uses terms such as ‘viability’ and ‘death’ with no clear outline of how these are defined. If ‘death’ is defined only as the result derived from an MTT assay, this does not tell us whether the result is truly correlated with a clonogenic assay.
This assay measures effects of photodamage on mitochondrial functions. After irradiation, the assay will be applied to untreated vs. treated cells. It is not clear (Fig. 3) how an assay can be used to determine whether the assay itself induces toxicity. During the assay, MTT will be converted to a colored product. Eventually, there will be no more to convert. This is what is shown in Figure 3. Cells are incubated with MTT and then subjected to an MTT assay? This makes no sense. The obvious method would be to use a clonogenic assay.
Summary:The MTT assay can be what is termed a ‘quick and dirty’ check on photodynamic cell damage. If the results are shown to be strongly correlated with clonogenic data, this approach can be used for some screening procedures. In the case of PDT, apoptosis is often a predominant death mechanism, mitochondria are often PDT targets, so the MTT assay will usually be appropriate. Moreover, the use of such an assay directly after irradiation will not (as discussed above) test for delayed or repairable photodamage. The authors discuss the need for controlling the cell number, the MTT concentration and time of incubations. Not mentioning the effect of varying the time between irradiation and carrying out the assay, along with the need for correlation with clonogenics, omits two potentially important determinants of relevance of this assay.
Minor point:
1. MTT in the title should be in all capital letters.
Author Response
please find attached the response to all questionings.

Reviewer 3 Report
Dear Authors,
the manuscript describes standardization and technical pitfalls of MTT assay in evaluation of photosensitizers cytotoxicity. In my opinion, the work does not show new informations and conclusions in this kind of study. The major point is that methylene blue is colorful, and in fact MTT assay should not be used with this kind of compounds. Furthermore, the Authors show results from testing only one compound. Also, it is well known that generally every protocol for any compound and cell line should be optimized and matched to the conditions of experiment. In my opinion, the manuscript is not suitable for publication in this journal.
Round 2
Reviewer 2 Report
This revised report discusses the potential use for the MTT assay in studies involving photodynamic therapy in cell culture. The report is designed to discuss causes of variability in assay results, not whether the assay truly reflects viability (although this is discussed). Important variables are the number of cells present, possible interference by photosensitizing agents, use of appropriate controls, etc. A few minor issues remain to be resolved.
1. Change Mtt to MTT in the title.
2. Check line 271, 273 and elsewhere: the method for identifying references for this journal does not require listing the names of authors in capital letters.
3. Line 93: change ‘cell’ to ‘cells’.
4. Line 91: adding Triton X-100, a nonionic detergent, will lyse cells and presumably inactivate mitochondrial dehydrogenase activity. Is there a suitable reference to indicate that this is true?
5. Line 125: the usual depiction of the Z’ factor shows the denominator as |μp - μn||.
6. Use of the term ‘viability’ in this report, including the figures, can be misleading since it is assumed that this is what is being measured. Lacking clonogenic assays, we can only be certain that the measurements reflect enzyme activity so some other term should be used. ‘Viability’ means the ability of a culture to proliferate and this is nowhere measured.
Reviewer 3 Report
Dear Authors,
unfortunately, I am upholding my previous comments and decision.
Author Response
Please, see attachment.

Round 3
Reviewer 3 Report
Dear Authors,
the manuscript may be a guide for researches which used MTT assay in their studies, especially that this assay is commonly used in PDT investigations.
Some typos in the text should be corrected.
